# Setting the Stage for Insulin Granule Dysfunction during Type-1-Diabetes: Is ER Stress the Culprit?

**DOI:** 10.3390/biomedicines10112695

**Published:** 2022-10-25

**Authors:** Aishwarya A. Makam, Anusmita Biswas, Lakshmi Kothegala, Nikhil R. Gandasi

**Affiliations:** 1Cell metabolism Lab (GA-08), Department of Molecular Reproduction, Development and Genetics (MRDG), Indian Institute of Science (IISc), Bengaluru 560012, India; 2Unit of Metabolic Physiology, University of Gothenburg, 405 30 Gothenburg, Sweden; 3Department of Medical Cell Biology, Uppsala University, BMC 571, 751 23 Uppsala, Sweden

**Keywords:** type-1-diabetes, ER stress, amyloids, autoantigens, insulin secretory granules

## Abstract

Type-1-diabetes (T1D) is a multifactorial disorder with a global incidence of about 8.4 million individuals in 2021. It is primarily classified as an autoimmune disorder, where the pancreatic β-cells are unable to secrete sufficient insulin. This leads to elevated blood glucose levels (hyperglycemia). The development of T1D is an intricate interplay between various risk factors, such as genetic, environmental, and cellular elements. In this review, we focus on the cellular elements, such as ER (endoplasmic reticulum) stress and its consequences for T1D pathogenesis. One of the major repercussions of ER stress is defective protein processing. A well-studied example is that of islet amyloid polypeptide (IAPP), which is known to form cytotoxic amyloid plaques when misfolded. This review discusses the possible association between ER stress, IAPP, and amyloid formation in β-cells and its consequences in T1D. Additionally, ER stress also leads to autoantigen generation. This is driven by the loss of Ca^++^ ion homeostasis. Imbalanced Ca^++^ levels lead to abnormal activation of enzymes, causing post-translational modification of β-cell proteins. These modified proteins act as autoantigens and trigger the autoimmune response seen in T1D islets. Several of these autoantigens are also crucial for insulin granule biogenesis, processing, and release. Here, we explore the possible associations between ER stress leading to defects in insulin secretion and ultimately β-cell destruction.

## 1. Introduction

Type-1-diabetes (T1D) is an autoimmune disorder caused by the inability of the pancreas to produce sufficient insulin, leading to high blood glucose levels. In most cases of T1D, the pancreas is targeted by the immune system, leading to selective destruction of the islet β-cells. This ultimately results in reduced insulin secretion and elevated blood glucose levels (hyperglycemia) [1]. T1D is also linked to α-cell dysfunction and glucagon overproduction, which aggravates hyperglycemia post food intake [2,3,4,5]. T1D is typically diagnosed in children and young adults (<15 years) [6]. It remains one of the most prevalent chronic illnesses diagnosed in childhood, affecting more than 1.2 million children and adolescents (0–19 years) [7]. T1D incidence is significantly associated with genetic predisposition, with multiple HLA Class I and II alleles leading to higher T1D risk. The prevalence of T1D is unevenly distributed across the world, with incidence rates being significantly higher in Europe and America [8] (Figure 1). The statistics presented in the 10th edition of the International Diabetic Federation Atlas (IDF) [7] estimated that T1D age-standardized incidences were highest in populations of northern European, Middle Eastern, and North African origin. The prevalence of T1D in Europe and America was 0.122%, while in Asia and Africa it was comparatively less, at 0.069% and 0.035%, respectively [9]. Studies of T1D in Indian populations are limited, due to the difficulty in diagnosis and distinction between T1D and T2D [10]. This is discussed further in the pathogenesis section.

## 2. A Concise Overview of the Pancreas

T1D is linked with dysregulated hormone secretions from the pancreas [1,11]. The pancreas is a 15–20 cm, leaf shaped organ located in the upper abdomen, behind the stomach in humans. Histologically, it comprises both exocrine and endocrine tissues [11]. The bulk of the pancreatic tissue, consisting of acinar cells, performs the exocrine function. The exocrine functions include the production of digestive enzymes for protein, carbohydrate, and fat digestion. Masked in a nest of acinar cells are patches of the endocrine tissue, the Islets of Langerhans. Rightly referred to as micro-organs, close to one million of these exist and cover about 1–2% of the total pancreatic mass. These islets include different cell types producing different hormones: (1) insulin producing β-cells, (2) glucagon producing α cells, (3) somatostatin producing δ cells, (4) pancreatic polypeptide producing F cells, and (5) ghrelin producing ε cells [11]. The proportions of the different cell types vary as a function of the islet age and size. Nearly 40–60% of the islets are β-cells [11]. The islets are nourished by one or more arterioles that branch into numerous capillaries, which later merge into small veins outside the islet [12].The architecture of the pancreas is affected during type-1- and type-2-diabetes [11]. One of the important characteristics of T1D is β-cell loss and islet amyloidosis, as depicted in Figure 2 [13].

## 3. Factors Influencing T1D Pathogenesis

Type-1-diabetes is a multifactorial disease, attributed to complex interactions between several genetic and environmental factors [14].

The genetic factors and their effects on the immune system have been analyzed with the help of Genome Wide Association Studies (GWAS), where more than 60 genes that play an important role in T1D pathogenesis have been discovered [15]. About 30–50% of the genetic risk factors identified are related to the Major Histocompatibility Complex (MHC) loci; mainly Human leukocyte antigens (HLA) class 2 alleles; HLA-DRB1, HLA-DQA1, and HLA-DQB1 [15]. The MHC patterns associated with increased risk of type-1-diabetes were found to be common in people of European descent [16,17]. GWAS studies have also found multiple non-MHC loci, such as the insulin (*INS)* loci associated with increased diabetes risk [17]. Along with these, some compensatory alleles have been characterized using single cell transcriptomic analysis [18]. The consequences of how they affect islet functions are discussed in the later part of this study.

The environmental factors involved in T1D may be both physical and physiological. The physical factors include (1) climate and (2) diet. Studies have discovered correlations between T1D incidence and regional climate [8]. It was observed that T1D incidence is higher in colder regions, particularly Canada, Australia, and the Nordic countries (Figure 1) [8]. Some dietary habits have also been correlated with increased T1D risk, namely diets with a high sugar and lactose intake [19]. Food intake affects the gut microbiome, and there is evidence showing possible correlations between the prevalence of T1D and the composition of the gut microbiome [20].

Physical factors such as climate and diet instigate viral infections such as enteroviral infections. This is an important physiological factor correlated with T1D pathophysiology. Enteroviruses such as CVBs show tropism towards β-cells [21]. The most direct evidence was the detection of CVB-specific RNA in the serum of 40–60% of recently diagnosed patients with insulin-dependent diabetes [22,23]. Viral strains have widely differing degrees of cytopathic effect. Several cytolytic viral strains can induce β-cell death in human islets, accompanied by morphological changes and impaired neogenesis of β-cells. In fact, it was observed that [24] pancreatic islets infected with the CVB-E2 strain failed to regenerate β-cells, and progressively accumulated dead tissue. This might be because viral infections can modulate multiple β-cell signaling pathways, leading to loss of cellular homeostasis, inflammation, and cell death [25]. On the contrary, there is evidence that viral infections prevent T1D pathogenesis in animal models, possibly by modulating the immune response [26]. For instance, a protective β-cell environment can be induced by causing the conversion of autoreactive T-cells to anti-inflammatory T_reg_ cells [26,27].

Multiple signaling pathways connected to ER stress are also selectively upregulated, aiding in viral replication [25,28]. Host translation machinery is hijacked for viral protein synthesis, which leads to a decrease in the synthesis of host proteins. Genes associated with insulin synthesis and release are downregulated in CVB infected β-cells, affecting the secretory ability of β-cells [25,28]. Viral infections induce pro-inflammatory cytokine production from β-cells [28]. This creates an inflammatory microenvironment and guides the macrophage and T-cell attack on β-cells. Immune cell attack on β-cells are further aggravated by the increased expression of MHC Class-I molecules in T1D individuals [29]. Loss of self-recognition may also occur as a result of molecular mimicry by viral proteins. For instance, the rotavirus VP7 protein mimics the IA-2/ICA512, an intrinsic protein of insulin granules. This leads to false recognition of a β-cell protein as a viral protein by CD8+ T-cells, leading to the destruction of β-cells [30]. In summary, the consequences of viral infection produce an inflammatory microenvironment and cause alterations in protein epitopes, leading to autoantigen generation, both of which can lead to T1D [31].

The factors discussed above lead to β-cell destruction via autoimmune attack. Autoimmunity results in an inflammatory environment, leading to cell death, as shown in Figure 2. The other major mechanism of β-cell destruction is via ER stress. The consequences of ER stress on β-cell destruction are discussed in detail in the following paragraphs.

## 4. ER stress and Its Consequences in T1D

Among all the membrane bound organelles of a eukaryotic cell, the endoplasmic reticulum (ER) is one of the most essential one. The ER is responsible for the proper folding of membrane and secretory proteins, synthesis of sterols and lipids, and a reserve for free Ca^++^ [32]. The lumen of the rough ER, which gets its title from the numerous ribosomes attached to its surface, provides an optimal microenvironment for protein processing [32]. Cells need optimal ER functioning to maintain homeostasis. The demand for functional proteins is higher in secretory cells, such as the islet β-cells. The β-cells have a basal level of translation required for cellular maintenance. In addition to this, they must process large amounts of hormones or enzymes rapidly in response to physiological cues [33]. This leads to an additional burden on the ER, resulting in ER stress. Apart from this, other external factors such as viral infection and exposure to toxins also result in ER stress. Other studies have shown that there is a close association between ER stress and ROS production. Overall, these factors cause a drop in the efficiency of ER function, leading to the accumulation of mutated, unfolded, or misfolded proteins [33,34]. ER stress could be characterized as a disequilibrium between the demand for functional proteins and the ability of the ER to fold the proteins [35]. Specialized signaling pathways are activated in response to ER stress. The most well-characterized facet of ER stress signaling is the unfolded protein response (UPR) [36]. ER stress engages the adaptive UPR pathway, which upregulates the production of ER chaperones and causes the phosphorylation of the α subunit of translation initiation factor 2 (eIF2α), to temporarily repress global translation [37,38,39,40]. If these stress response pathways are unable to restore proteostasis, which leads to activation of terminal UPR, which upregulates pro-apoptotic factors, such as CHOP/GADD153, and drives the cell to apoptosis [37]. Cell injury due to chronic ER stress is being increasingly recognized as a common contributor to diabetes [41].

The connection between ER stress and T1D pathogenesis was first drawn from the observation that ER stress markers were upregulated in T1D β-cells compared to healthy β-cells. A direct link for β-cell ER stress and T1D was then provided by showing that T1D pathogenesis could be halted by mitigating β-cell ER stress [42]. The major result of ER stress in β-cells are the hurdles in protein processing. One well-studied example is the islet amyloid polypeptide (IAPP or amylin), a peptide hormone co-secreted with insulin. Pro-islet amyloid polypeptide (proIAPP) is produced as a 67 amino acid pro-peptide and undergoes post-translational modifications, including protease cleavage in the Golgi complex and insulin secretory granules, to produce mature IAPP [43,44,45]. Mature IAPP is stored in the insulin secretory granule and is found in the halo region of the granule, while insulin is located in the dense core [45]. Physiological levels of IAPP are approximately 1% those of insulin levels and are co-secreted during insulin exocytosis [43]. The IAPP released during insulin exocytosis plays a role in glycemic regulation by slowing gastric emptying and promoting satiety. Thereby, it prevents postprandial spikes in blood glucose levels [46].

## 5. IAPP and Amyloids

IAPP is characterized by its unique ability to aggregate and form toxic, insoluble, and pro inflammatory amyloid aggregates [43,45]. These aggregates are present in the majority of individuals with T2D and have been widely studied in T2D pathophysiology [47,48]. The possible role of IAPP amyloids in T1D pathology remains relatively unexplored in large sample sets. A study from the Better Diabetes Diagnosis cohort revealed that ~11% of patients with T1D had markedly elevated plasma IAPP levels relative to C-peptide and pro-insulin levels [48]. The accumulation of IAPP amyloid aggregates in β-cells induces ER stress, triggering endoplasmic reticulum-associated protein degradation (ERAD) and an unfolded protein response (UPR), as discussed before.

This has also been investigated at the single cell level, using cell line models (INS1E, etc.) by overexpressing human IAPP (hIAPP), where ER stress causes IAPP cytotoxicity [49]. ER stress is artificially induced in these cell lines using chemicals such as thapsigargin. Glucose-stimulated insulin secretion of INS1E cells having higher IAPP levels was comparable to cells with physiological levels of IAPP [49]. However, induction of ER stress coupled with hIAPP overexpression at high glucose (16.7 mM) severely diminishes glucose-stimulated insulin release from hIAPP-INS1E cells compared with untreated control cells. Treatment with chemical chaperones that alleviate ER stress (measured using ER stress markers such as PERK, CHOP, etc.) can rescue this hIAPP-induced cytotoxicity [49]. Last year, it was found that TXNIP, a protein upregulated during ER stress, is a strong transcriptional activator of IAPP in islet cells [50]. Thus, ER stress may possibly trigger greater amyloid formation, by increasing IAPP levels. This may indicate a positive feedback loop, as depicted in Figure 3, that eventually induces apoptosis and autoimmune activity.

## 6. Loss of Ca^++^ Homeostasis as a Mechanism of Autoantigen Generation during ER Stress 

Another major consequence of ER stress is autoantigen generation [33]. A large percentage of T1D patients have characteristic autoantibodies against several β-cell proteins, acting as autoantigens [51]. A possible mechanism of autoantigen generation has been linked to the loss of calcium ion homeostasis that occurs during ER stress [32]. As mentioned earlier, the ER lumen is the largest reserve of Ca^++^ ions in the cell. ER transmembrane pumps are responsible for maintenance of Ca^++^ homeostasis [52]. During ER stress, the functioning of these pumps is affected, and Ca^++^ ions from the ER lumen enter the cytosol in large quantities. This leads to a spike in cytosolic Ca^++^ The loss of ionic homeostasis has many adverse effects on cellular signaling and enzyme activity. In particular, increased cytosolic Ca^++^ can cause abnormal activation of post-translational modification (PTM) enzymes and apoptosis. The activation of PTM enzymes and the role of PTMs has been studied in various autoimmune diseases, such as rheumatoid arthritis [53], multiple sclerosis [54], etc. Activation of the PTM enzymes such as peptidyl arginine deiminase (PAD) and Tissue transglutaminase 2 (Tgase2), specifically during ER stress is triggered by the influx of calcium ions into the cytoplasm. Their enzyme activity has also been linked to the generation of autoantigens in T1D [33,55,56]. These enzymes, upon activation, can modify constitutive β-cell proteins such as chromogranin A [57,58], preproinsulin [59], TSPAN7 [60,61], GAD65 [62,63], and ZnT8 [64], etc. Thus, the loss of calcium homeostasis during ER stress leads to aberrant post-translational modifications of β-cell proteins. These modifications alter the tertiary structure of peptides and produce autoantigenic epitopes that generate an autoantibody response.

## 7. Effects of ER Stress on Large Dense Core Vesicles—Insulin Secretory Granules (ISGs)

Large dense core vesicles (LDCVs) are vesicles specialized for the storage of secretory proteins. LDCVs are formed in the trans-Golgi network (TGN), where specific pH and redox conditions prevail. The core of these vesicles is dense and contains large amounts of granulogenic protein aggregates. β-cells produce many such LDCVs for insulin storage and secretion. These are known as insulin secretory granules (ISGs) [65,66] (Figure 4). A fraction of these granules is trafficked to the plasma membrane to prepare them for release. An elaborate process of docking and priming precedes the fusion of these vesicles with the membrane [67,68,69,70], ultimately releasing insulin upon the increase in cytoplasmic Ca^++^ [71]. Many of the proteins involved in this process are similar to the ones involved in synaptic transmission [72]. More recently, proteins that are not found in synaptic transmission have been identified as unique players [67,72]. These have been shown to be involved in β-cell secretion. This underscores the importance of the further investigations required in this area. Some of these players have a huge relevance in ER stress and the development of T1D [58,73]. Proteins such as TSPAN7 and ZnT8, involved in trafficking of insulin granules, are also known to act as autoantigens during T1D. ZnT8 was first identified as a major T1D autoantigen by exploiting the bioinformatics approaches on β-cell-specific proteins [64,74]. The presence of ZnT8 autoantibodies has been further documented in 60–80% of Caucasian patients [64] with type-1-diabetes, and in 58% of patients in the Japanese population [75]. ZnT8 is a key member of the insulin secretory granules (ISGs) and is mainly responsible for the intake of zinc ions into the granules [75] (Figure 4). ZnT8 may also regulate the function of proinsulin processing enzymes such as Pro convertase 1/2 (PC1/2), which have zinc ions as a cofactor [64]. Zinc ions stabilize the stored form of insulin and mediate the release of insulin from secretory granules [74,75]. It is likely that recognition of ZnT8 as an autoantigen on ISG membranes has a negative impact on insulin secretion and processing during T1D (Figure 4). The binding of autoantibodies to ZnT8 may also affect its function as an ion transporter, leading to a deficiency of zinc in insulin secretory granules. It has previously been observed that knock outs of ZnT8 show a loss of the dense core of Zn-insulin crystals, which results in abnormalities in glucose tolerance, insulin processing, and secretion [74,76]. Impaired functioning of ZnT8, due to recognition as a T1D autoantigen, may thus promote β-cell dysfunction, insulin release, and an inflammatory microenvironment.

Tetraspanins are transmembrane molecules that act as protein organizers and play a role in vesicle trafficking in the neurons. Tetraspanin 7 expression was also detected in the pancreatic islets [77] (Figure 4). β-cells, being secretory in nature are enriched with L-type Ca^++^ channels. These Ca^++^ channels are important in the glucose stimulated insulin response (GSIS) [78]. Tetraspanin 7 has been shown to be involved in regulating the function of the L-type Ca^++^ channels of the β-cells [79]. In TSPAN7 knockdown (KD) cells, at a basal glucose concentration of 7 mM, a modest elevation in Ca^++^ concentration was observed. Insulin secretion is pulsatile [80] and it synchronizes with Ca^++^ oscillations. The frequency of these oscillations is increased in TSPAN7 KD cells. This disturbs the cellular homeostasis, since it is important for the β-cells to maintain a fine balance between insulin production and insulin secretion [65].

The production of insulin is regulated by a *cis* element in the 5′ UTR of the preproinsulin mRNA (ppIGE). This regulation occurs in a glucose-dependent manner [81]. Glucose stimulation enhances proinsulin translation tenfold [81]. This imposes a burden on the ER of the insulin-producing β-cells. Supportive of this view, it has also been shown that the reduction of insulin production alleviates ER stress [49,82,83]. Given that ER functioning is also needed for the processing and folding of proinsulin [82,84], ER stress results in altered forms of insulin that are autoantigenic [55,85]. The recognition of insulin as an autoantigen further leads to inflammation, cytokine release, and an autoimmune response, culminating in T1D [86].

## 8. Conclusions

ER stress is predominant in the β-cells of T1D patients. Major repercussions of ER stress are defects in protein processing and autoantigen generation. A feedback loop sets in, where ER stress raises the levels of misfolded proteins, such as IAPP. Misfolded IAPP, aggregates to form amyloids, which can, in turn, aggravate pre-existing ER stress. There are several other proteins showing defects in processing. Examples include TSPAN7 and ZnT8, which play a role in the biogenesis and trafficking of ISGs. Notably, these proteins have also been characterized as T1D autoantigens. This suggests the involvement of ER stress in autoantigen generation. The scope of future research lies in investigating the specific trafficking proteins that are affected due to ER stress, so that the consequences for insulin release can be reversed. The specific autoantigens leading to inflammatory response can be countered, as possible treatment strategies for type-1 diabetics.

## Figures and Tables

**Figure 1 biomedicines-10-02695-f001:**
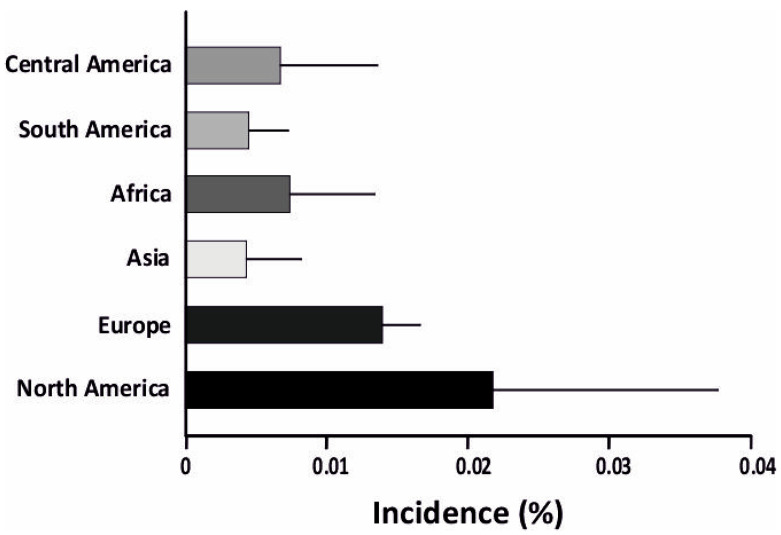
Incidence of T1D is higher in North America and Europe compared to other parts of the world. Notably, Asia has the lowest incidence of T1D. The incidence is plotted as a percentage, showing the incidence of T1D.

**Figure 2 biomedicines-10-02695-f002:**
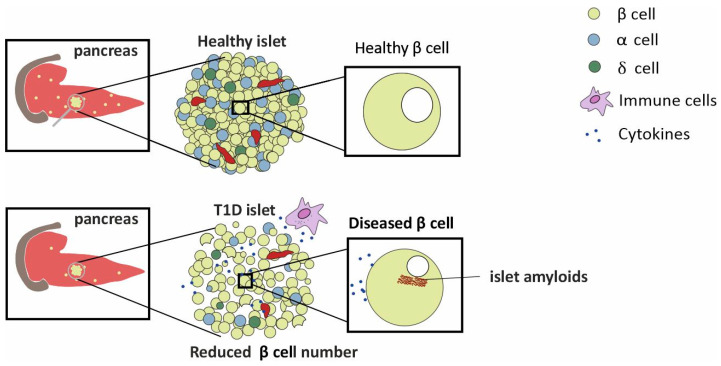
The difference in the number of β-cells in healthy and T1D islets is depicted. A diseased islet shows the presence of amyloids.

**Figure 3 biomedicines-10-02695-f003:**
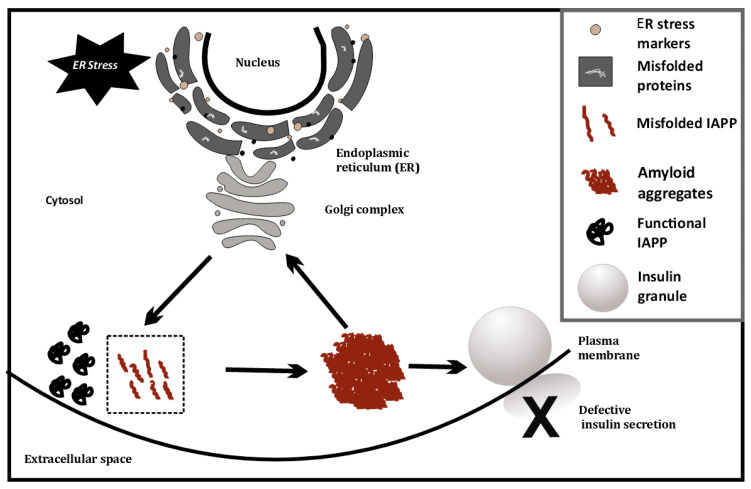
ER stress-induced misfolding of IAPP in β-cells, leading to amyloid formation.

**Figure 4 biomedicines-10-02695-f004:**
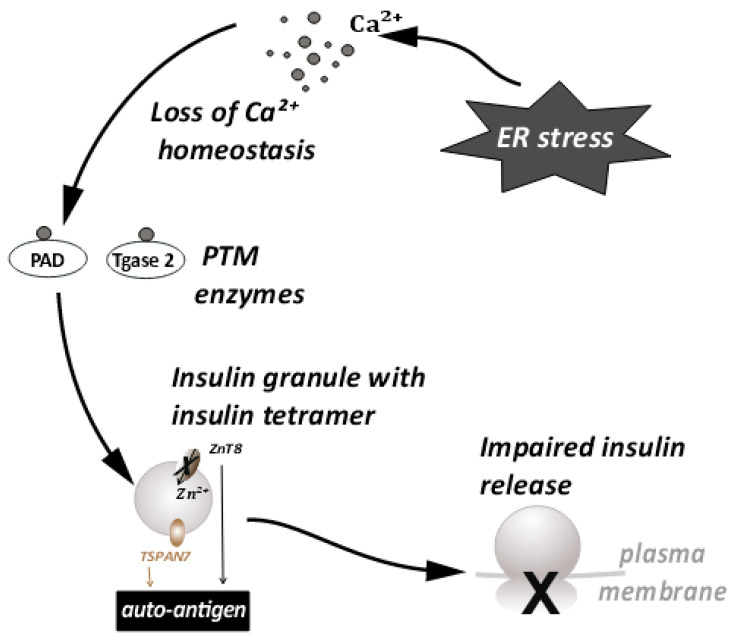
ER stress disturbs the Ca^++^ ion homeostasis. This leads to the conversion of proteins important for insulin granule processing and release into autoantigens. This further leads to impaired insulin release.

## Data Availability

The datasets used in the study have previously been published and are readily available.

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
