# Peer review of "Setting the Stage for Insulin Granule Dysfunction during Type-1-Diabetes: Is ER Stress the Culprit?"

_biomedicines, 2022, doi:10.3390/biomedicines10112695_

Round 1
Reviewer 1 Report
In the review, Makam et al explore the topics of ER stress and islet amyloid formation as factors that contribute to the development of type 1 diabetes. In general the article is well organized and accurate, but the authors should consider the following points:
1) The first sentence of the introduction is inaccurate - in T1D the pancreas is not unable to produce insulin (it is very rare at diagnosis for patients to have no detectable c-peptide). Rather, insulin production is reduced to levels that are insufficient to control blood glucose levels.
2) Section 1 mentions geographic disparity in the incidence of T1D. Although GWAS are discussed later in the article, it may be helpful to briefly mention genetics in that paragraph.
3) The topic of viral pathogenesis (as articulated in section 3) remains controversial. The authors should add a sentence to acknowledge that fact. The activity of viruses may not be required to initiate the feed-forward cycle of ER stress that leads to dysfunction, increased demand, and additional ER stress.
4) The are two distinct UPR pathways - the adaptive UPR and the terminal UPR. It would be good to mention that distinction.
5) The conclusion does an adequate job of summarizing the main ideas of the article, but it would be better to also include a few forward looking statements about important unanswered questions and/or ideas to translate this mechanistic knowledge to treat the disease.
6) The content of Figure 2 seems inadequate. Please include a little more detail about differences between the pancreas, islets, and beta cells in healthy versus T1D subjects. For example, it could be shown that T1D islets have immune infiltrates and that diseased beta cells can be under stress.
7) A few general linguistic edits are needed. For example, the terms "relations" (line 21), "blotches" (line 55), and "a huge" (line 61) seem nonstandard.
Author Response
Reviewer -1
In the review, Makam et al explore the topics of ER stress and islet amyloid formation as factors that contribute to the development of type 1 diabetes. In general the article is well organized and accurate, but the authors should consider the following points:
1) The first sentence of the introduction is inaccurate - in T1D the pancreas is not unable to produce insulin (it is very rare at diagnosis for patients to have no detectable c-peptide). Rather, insulin production is reduced to levels that are insufficient to control blood glucose levels.
Thank you for pointing this out. We have corrected this. Now it reads as “Type-1-diabetes (T1D) is an autoimmune disorder caused by the inability of the pancreas to produce sufficient insulin, leading to high blood glucose levels.”.
Lines – 53-54 (in the version with track changes)
Lines – 34-35 (in the final version)
2) Section 1 mentions geographic disparity in the incidence of T1D. Although GWAS are discussed later in the article, it may be helpful to briefly mention genetics in that paragraph.
Appreciate the thought there, we have added a sentence “T1D incidence is significantly associated with genetic predisposition, with multiple HLA class I and II alleles leading to higher T1D risk.” highlighting this. Thank you.
Lines – 63-64 (in the version with track changes)
Lines – 42-43 (in the final version)
3) The topic of viral pathogenesis (as articulated in section 3) remains controversial. The authors should add a sentence to acknowledge that fact. The activity of viruses may not be required to initiate the feed-forward cycle of ER stress that leads to dysfunction, increased demand, and additional ER stress.
This was a very important and useful suggestion, thank you. We have added the clarifications required. We have included a few lines as follows “On the contrary there is evidence that viral infections have been shown to prevent T1D pathogenesis in animal models, possibly by modulating the immune response. For instance, protective β-cell environment can be induced by causing the conversion of autoreactive T-cells to anti-inflammatory Treg cells. “ .
Lines – 131-134 (in the version with track changes)
Lines – 111-114 (in the final version)
4) The are two distinct UPR pathways - the adaptive UPR and the terminal UPR. It would be good to mention that distinction.
Thanks for the suggestion. We have added an explanation in the new version in the following sentences “ER stress engages the adaptive UPR pathway, which up-regulates the production of ER chaperones and causes the phosphorylation of the α subunit of translation initiation factor 2 (eIF2α) to temporarily repress global translation. If these stress response pathways are unable to restore proteostasis, it leads to activation of terminal UPR, which upregulates pro-apoptotic factors such as CHOP/GADD153, and drives the cell to apoptosis. Cell injury due to chronic ER stress is being increasingly recognized as a common contributor to diabetes.”.
Lines – 174-181 (in the version with track changes)
Lines – 153-159 (in the final version)
5) The conclusion does an adequate job of summarizing the main ideas of the article, but it would be better to also include a few forward looking statements about important unanswered questions and/or ideas to translate this mechanistic knowledge to treat the disease.
This is a valuable suggestion which we have incorporated into our conclusion, along with a few other changes. The present conclusion is as follows“ ER stress is a prominent feature in the β-cells of T1D patients. Major repercussions of ER stress are defects in protein processing and autoantigen generation. A positive feedback loop sets in where ER stress raises the levels of misfolded proteins, like IAPP. Misfolded IAPP aggregates to form amyloids which can in turn aggravate the pre-existing ER stress. There are several other proteins showing defects in processing. Examples include TSPAN7 and ZnT8, which play a role in the biogenesis and trafficking of ISGs. Notably, these proteins have also been characterized as T1D autoantigens. Here, we suggest the involvement of ER stress in autoantigen generation. The future scope lies in investigating the specific trafficking proteins that are affected due to ER stress so that the consequences on the insulin release can be reversed. The specific autoantigens leading to inflammatory response can be countered as possible treatment strategies for type-1 diabetics.
Lines – 329-339- page break (in the version with track changes)
Lines – 285-295-page break (in the final version)
6) The content of Figure 2 seems inadequate. Please include a little more detail about differences between the pancreas, islets, and beta cells in healthy versus T1D subjects. For example, it could be shown that T1D islets have immune infiltrates and that diseased beta cells can be under stress.
We have made the necessary changes to highlight these aspects mentioned. Thank you for the suggestion.
Lines – 94-97 (in the version with track changes)
Lines – 74-77 (in the final version)
7) A few general linguistic edits are needed. For example, the terms "relations" (line 21), "blotches" (line 55), and "a huge" (line 61) seem nonstandard.
This has been corrected. We have changed the words to “association” (line 25- in the version with track changes, line 22 – in the final version), “patches” (line 84- in the version with track changes, line 64 – in the final version )and we have removed “a huge” from line. (line 90- in the version with track changes, line 70 – in the final version)

Reviewer 2 Report
Title: Cellular determinants of type-1-diabetes: Spotlight on ER stress and islet amyloids leading to defects in insulin release
Manuscript Number:
This manuscript focused on type 1 diabetes-associated endoplasmic reticulum (ER) stress in pancreatic β cells, and reviewed the interaction of ER stress with factors such as protein folding defects, vesicle release and loss of ion homeostasis during endoplasmic reticulum stress. I have some comments which should be addressed before the final decision by the journal:
1. Abstract: This section should be rewritten. How do you know that the global prevalence of type 1 diabetes (T1D) is 0.05%? Would it be appropriate for the sample statement on lines 23-25 to appear in this section? The impact of environmental and cellular factors such as protein folding defects, vesicle release, and loss of ion homeostasis during ER stress on pancreatic β-cell survival associated with T1D development should be clearly and concisely addressed.
2. Part 3 (Line 67 -): The presentation of this part is not clear and does not summarize. Could autoimmunity or stress-induced destruction of pancreatic β cells under the complex interplay of genetic and environmental factors be the mechanism of type 1 diabetes? This part does not mention the important role of pancreatic β cells destruction in type 1 diabetes, which is out of touch with the following text. Rather than describing the pathogenesis of type 1 diabetes, it is more about listing its influencing factors.
3. Part 4 (Line 116-): Lines 123-128, there is a problem with the statement that it is not only the increased demand for protein synthesis that causes ER stress in islet beta cells, such as viral infection, chemical and toxin exposure, reactive oxygen species ( ROS) is also possible; lines 144-150, the description is not clear, is it to show that endoplasmic reticulum stress can affect the synthesis of IAPP and proIAPP?
4. Part 5 (line 152-): Why do you mention the cytotoxicity of proIAPP in the first sentence, isn't the subject heading of this part IAPP and amyloids?
5. Part 6 (line 179-): The statement is not clear, it is to explain that the production of autoantigens related to calcium homeostasis involved in endoplasmic reticulum stress, which affects the survival of pancreatic β cells?
6. Part 7 (line 200-): The description is not clear, it can be combined with Part 6, according to Figure 4, it illustrates the calcium homeostasis-related autoantigens and insulin secretion granules involved in endoplasmic reticulum stress production. Ion homeostasis (calcium ions) plays an important role and should be highlighted.
7. Part 8 (Line 256): Figure 1 is not aesthetically pleasing.
8. Summary (Line 272-): Reading through the full text does not match the title. Are the authors trying to show that ER stress-induced amyloid formation and beta cell autoantigen production affects patients with type 1 diabetes?

Author Response
This manuscript focused on type 1 diabetes-associated endoplasmic reticulum (ER) stress in pancreatic β cells, and reviewed the interaction of ER stress with factors such as protein folding defects, vesicle release and loss of ion homeostasis during endoplasmic reticulum stress. I have some comments which should be addressed before the final decision by the journal:
1. Abstract: This section should be rewritten. How do you know that the global prevalence of type 1 diabetes (T1D) is 0.05%? Would it be appropriate for the sample statement on lines 23-25 to appear in this section? The impact of environmental and cellular factors such as protein folding defects, vesicle release, and loss of ion homeostasis during ER stress on pancreatic β-cell survival associated with T1D development should be clearly and concisely addressed.
We had initially taken the prevalence data from the International Diabetes foundation (IDF) Atlas, 2021. But given a more recent study published in Lancet by Gregory et al, 2022, we have accordingly updated the number.
Lines – 16-17(in the version with track changes)
Lines – 13-14 (in the final version)
Thanks for the suggestions. We have made several changes and rewritten the abstract and have also addressed the concerns raised in lines 23-25. The abstract has been changed to “Type-1-diabetes (T1D) is a multifactorial disorder with a global incidence of about 8.4 million individuals worldwide in 2021. It is primarily classified as an autoimmune disorder, where the pancreatic β-cells are unable to secrete sufficient insulin. This leads to elevated blood glucose levels (hyperglycemia). The development of T1D is an intricate interplay between various risk factors, such as genetic, environmental and cellular elements. In this review we focus on the cellular elements such as ER (endoplasmic reticulum) stress and its consequences on T1D pathogenesis. One of the major repercussions of ER stress is defective protein processing. A well-studied example is that of  Islet amyloid polypeptide (IAPP) which is known to form  cytotoxic amyloid plaques when misfolded. This review discusses the possible association between ER stress, IAPP and amyloid formation in β-cells and its consequences in T1D. Another repercussion of ER stress is autoantigen generation. This is driven by the loss of Ca2+ ion homeostasis. Imbalanced Ca2+ levels lead to abnormal activation of enzymes causing post-translational modification of β-cell proteins. These modified proteins act as autoantigens and trigger an autoimmune response seen in T1D islets. Several of these autoantigens are also crucial for insulin granule biogenesis, processing and release. Here we explore the possible associations between ER stress leading to defects in insulin secretion and ultimately β-cell destruction.”.
Lines – 13-30 (in the version with track changes)
Lines – 16-33 (in the final version)
2. Part 3 (Line 67 -): The presentation of this part is not clear and does not summarize. Could autoimmunity or stress-induced destruction of pancreatic β cells under the complex interplay of genetic and environmental factors be the mechanism of type 1 diabetes? This part does not mention the important role of pancreatic β cells destruction in type 1 diabetes, which is out of touch with the following text. Rather than describing the pathogenesis of type 1 diabetes, it is more about listing its influencing factors.
Thank you for pointing this out. We have changed the title of the paragraph to “Factors influencing pathogenesis of T1D “to suit the content presented.
Line – 98 (in the version with track changes)
Lines – 78 (in the final version)
We have also added a few lines on how all of these influencing factors culminate β cell destruction to summarize it all. The following are the lines added “The factors discussed above leads to β-cell destruction via autoimmune attack. Autoimmunity results in an inflammatory environment, leading to cell death as shown in Figure 2. The other major mechanism of β-cell destruction is via ER stress. The consequences of ER stress on β-cell destruction are discussed in detail in the following paragraphs.”.
Lines – 149-153 (in the version with track changes)
Lines – 129-133 (in the final version)
3. Part 4 (Line 116-): Lines 123-128, there is a problem with the statement that it is not only the increased demand for protein synthesis that causes ER stress in islet beta cells, such as viral infection, chemical and toxin exposure, reactive oxygen species (ROS) is also possible; lines 144-150, the description is not clear, is it to show that endoplasmic reticulum stress can affect the synthesis of IAPP and proIAPP?
Thank you for bringing up this point. We have included the other factors that lead to ER stress in the following lines “Apart from this, other external factors such as viral infection, exposure to toxins also result in ER stress. Other studies have shown that there is a close association between ER stress and ROS production. Overall, these factors cause a drop in the efficiency of ER functioning leading to a freight of mutated, unfolded or misfolded proteins to accumulate.”.
Lines – 165-169 (in the version with track changes)
Lines – 145-149 (in the final version)
In lines 144-150, we just wanted to highlight the connection between ER stress and T1D pathogenesis. To support this connection, we have given a brief introduction about IAPP at the end of the paragraph. IAPP has remained our focus in the following paragraph, where we discuss in detail about the associations between ER stress and IAPP.
4. Part 5 (line 152-): Why do you mention the cytotoxicity of proIAPP in the first sentence, isn't the subject heading of this part IAPP and amyloids?
Sorry for the confusion here, the idea was to link the previous paragraph here. We have incorporated your suggestion and removed the sentence “ProIAPP and partially processed proIAPP may cause cytotoxicity.” for better clarity.
5. Part 6 (line 179-): The statement is not clear, it is to explain that the production of autoantigens related to calcium homeostasis involved in endoplasmic reticulum stress, which affects the survival of pancreatic β cells?
We appreciate the reviewers’ suggestions in this part.
Yes. We have modified the title for better clarity into “Loss of Ca++ homeostasis as a mechanism of autoantigen generation during ER stress.”
Lines – 230 (in the version with track changes)
Lines – 204 (in the final version)
We have also added few sentences in the paragraph to improve the understanding of this part. The modified paragraph reads as “Another major fallout of ER stress is autoantigen generation. A large percentage of T1D patients have characteristic autoantibodies against several β-cell proteins, acting as autoantigens. A possible mechanism of autoantigen generation has been linked to the loss of calcium ion homeostasis that occurs during ER stress. As mentioned earlier, the ER lumen is the largest reserve of Ca++ ions in the cell. ER transmembrane pumps are responsible for maintenance of Ca++ homeostasis. During ER stress, the functioning of these pumps is affected and Ca++ ions from ER lumen enter the cytosol in large quantities. This leads to a spike in cytosolic Ca++. The loss of ionic homeostasis has many adverse effects on cellular signaling and enzyme activity. In particular, increased cytosolic Ca++ can cause abnormal activation of post translational modification (PTM) enzymes and apoptosis. The activation of PTM enzymes and the role of PTMs has been studied in various autoimmune diseases, such as rheumatoid arthritis, multiple sclerosis etc. Activation of the PTM enzymes like peptidyl arginine deiminase (PAD) and Tissue transglutaminase 2 (Tgase2) specifically during ER stress is triggered by the influx of calcium ions into the cytoplasm. Their enzyme activity has also been linked to the generation of autoantigens in T1D. These enzymes upon activation can modify constitutive β--cell proteins like chromogranin A, preproinsulin, TSPAN7, GAD65 and ZnT8 etc. Thus, the loss of calcium homeostasis during ER stress leads to aberrant post-translational modifications of β-cell proteins. The modifications alter the tertiary structure of peptides and produce autoantigenic epitopes that generate an autoantibody response.”.
Lines – 232-252 (in the version with track changes)
Lines – 204-224 (in the final version)
6. Part 7 (line 200-): The description is not clear, it can be combined with Part 6, according to Figure 4, it illustrates the calcium homeostasis-related autoantigens and insulin secretion granules involved in endoplasmic reticulum stress production. Ion homeostasis (calcium ions) plays an important role and should be highlighted.
Thank you for this suggestion. We have altered certain sentences which would now clearly highlight the importance of Ca2+ ion homeostasis in the previous paragraph (Part 6). This has been addressed along with the changes in comment 5.
7. Part 8 (Line 256): Figure 1 is not aesthetically pleasing.
We have redone figure 1 to make it look better. Thanks for pointing this out.
Lines – 53-57 (in the version with track changes)
Lines – 72-76 (in the final version)
8. Summary (Line 272-): Reading through the full text does not match the title. Are the authors trying to show that ER stress-induced amyloid formation and beta cell autoantigen production affects patients with type 1 diabetes?
We have modified our title to better match our content into “Setting the stage for insulin granule dysfunction during Type-1-diabetes: is ER stress the culprit? “.
Lines – 2-3 (in the version with track changes)
Lines – 2-3 (in the final version)
